# Simultaneous Maturation of Single Chain Antibody Stability and Affinity by CHO Cell Display

**DOI:** 10.3390/bioengineering9080360

**Published:** 2022-08-02

**Authors:** Ruiqi Luo, Baole Qu, Lili An, Yun Zhao, Yang Cao, Peng Ren, Haiying Hang

**Affiliations:** 1Key Laboratory for Protein and Peptide Pharmaceuticals, National Laboratory of Biomacromolecules, Institute of Biophysics, Chinese Academy of Sciences, Beijing 100101, China; 18810991228@163.com (R.L.); yongheng49@163.com (B.Q.); afxdxz@163.com (L.A.); super@moon.ibp.ac.can (Y.Z.); 2University of Chinese Academy of Sciences, Beijing 100049, China; 3Center of Growth, Metabolism and Aging, Key Laboratory of Bio-Resources and Eco-Environment of Ministry of Education, College of Life Sciences, Sichuan University, Chengdu 610064, China; 4Department of Thoracic Surgery, Peking University Third Hospital, Beijing 100191, China

**Keywords:** stability, affinity maturation, mammalian cell display, thermo-resistant CHO cells

## Abstract

Antibody stability and affinity are two important features of its applications in therapy and diagnosis. Antibody display technologies such as yeast and bacterial displays have been successfully used for improving both affinity and stability. Although mammalian cell display has also been utilized for maturing antibody affinity, it has not been applied for improving antibody stability. Previously, we developed a Chinese hamster ovary (CHO) cell display platform in which activation-induced cytidine deaminase (AID) was used to induce antibody mutation, and antibody affinity was successfully matured using the platform. In the current study, we developed thermo-resistant (TR) CHO cells for the purpose of maturing both antibody stability and affinity. We cultured TR CHO cells displaying an antibody mutant library and labeled them at temperatures above 41 °C, enriching cells that displayed antibody mutants with both the highest affinities and the highest display levels. To evaluate our system, we chose three antibodies to improve their affinities and stabilities. We succeeded in simultaneously improving both affinities and stabilities of all three antibodies. Of note, we obtained an anti-TNFα antibody mutant with a Tm (dissolution temperature) value 12 °C higher and affinity 160-fold greater than the parent antibody after two rounds of cell proliferation and flow cytometric sorting. By using CHO cells with its advantages in protein folding, post-translational modifications, and code usage, this procedure is likely to be widely used in maturing antibodies and other proteins in the future.

## 1. Introduction

Antibodies are important tools for a broad range of applications. However, to be practically useful, antibodies must be stable and bind their target antigens with sufficiently high affinity. Stability is an important antibody feature, critical for expression, activity, specificity, and storage [1,2,3,4,5,6]. Stabilization engineering of antibodies can be accomplished in two completely different ways. One is through directed evolution, and the other is through the rational design of mutants for engineering modification. The latter needs to be based on certain structural analysis and rational design to obtain antibody mutants with high stability, then after expression purification and stability testing, to verify whether the antibody is thermally stable. The two strategies may compensate for each other. The rational design is quick to perform and may consider detailed intramolecular bonding mechanisms such as forming new disulfide bonds. The evolution approach uses out-of-the-box thinking, often resulting in enhanced stability in surprising ways. In vivo maturation is used to increase both affinity and stability of antibodies [7]. In recent years, a few approaches to improving antibody thermodynamic stability have been described, including phage display [8], bacterial display [9], and yeast display (Traxlmayr and Obinger 2012). Among them, yeast cell display is most often used to evolve antibody stability. In vitro mammalian cell display has been utilized to characterize the stability of antibody mutant clones [10,11] but has not been used to evolve an antibody for improved stability. One of the reasons is that mammalian cells are not as heat-resistant as bacteria and yeast, so a heat challenging procedure cannot be applied to mammalian cell display.

In this study, we developed thermo-resistant (TR) CHO cells and simultaneously matured antibody stability and affinity by culturing and labeling cells displaying an antibody mutant library at temperatures above 37 °C, followed by flow sorting and collecting TR CHO cells that displayed high levels of antibodies with the highest affinities. We applied the above procedure to three different antibodies and obtained a series of mutant antibodies with significant improvement in their affinities and stabilities.

## 2. Materials and Methods

### 2.1. Plasmid Construction and Expression

TNFα-GFP is His-tagged on its N-terminus and was prepared by following the procedure by Chen et al. [12]. Briefly, His-TNFα-GFP was synthesized in *E. coli* containing the PET28a (+)-TNFα-GFP plasmid and was purified with a Ni column. The concentration of the prepared TNFα-GFP was 2 mg/mL. CD19-GFP and human kappa CL (light chain constant region) -GFP were synthesized in expi293F cell and purified with a Ni column.

Anti-human TNFα wild-type scFv (αTNFα) is from Jie Tang (Institute of Biophysics, Chinese Academy of Sciences, Beijing, China). The dual recombinase expression plasmid pCI-Flp-2A-Cre for co-expression of Flpo and iCre recombinases, pFRT-αTNFα-HA-loxP for the replacement of the puro with the antibody (TNFα antibody) gene in a predetermined genomic locus of CHO-puro cells were described in detail previously [13]. pFRT-αTNFα-HA-loxP was created by replacing SP-anti-TNFα single chain antibody gene-HA-TM (SP: signal peptide; HA: HA tag; TM: transmembrane) with SP- HA-anti-TNFα single chain antibody gene-TM [13] between *Bam*HI and *Bgl*II in pFRT-αTNFα-loxP. Briefly, Flpo and iCre cut puro gene in the CHO chromosome and inserted αTNFα-HA into the location. Similarly, pFRT-anti CD19 scFv-HA-loxP was created by replacing SP-anti CD19 scFv gene-HA-TM with SP-HA-anti-TNFα single chain antibody gene-TM, and pFRT-anti-human kappa chain of CL-HA-loxP was created by replacing SP-anti-human kappa chain of CL gene-HA-TM with SP-HA-anti-TNFα single chain antibody gene-TM.

The DNA sequences of mAID-plus were synthesized by the Genscript Biotech Company (Nanjing, China). The PCR primer pairs for the mAID-plus contained *Hin*dIII and *Xho*I restriction sites, and the PCR product was cut with *Hin*dIII and *Xho*I. The purified fragment was inserted between the *Hin*dIII and *Xho*I restriction sites in pCEP4-Ig-Ek plasmid.

Anti TNFα single chain antibody, anti CD19 scFv, and anti-human kappa chain of CL wild-type and mutants with his tag on C-terminus were expressed in 293F cells and purified with a Ni column.

### 2.2. Cell Culture

CHO/dhFr^-^ cells (12200036, Cell Bank of the Chinese Academy of Sciences, Shanghai, China) and cell lines derived from CHO/dhFr were propagated in IMDM medium (HyClone, Logan, UT, USA) containing 10% fetal bovine serum (HyClone, Logan, UT, USA), 0.1 mM hypoxanthin, and 0.016 mM thymidine (HT, Gibco, Waltham, MA, USA), at 37 °C in a 5% CO_2_ incubator. Subcultures were carried out every 2–3 days. CHO-puro cells (The CHO/dhFr cells with a single copy of retargetable high-level expression cassette) were established previously and described in detail [13]. The suspension Expi293F cells (Invitrogen, Carlsbad, CA, USA) were cultured in SMM 293-TII medium (Sino Biological Inc., Beijing, China) in suspension at 37 °C, 5% CO_2_, 125 rpm. Cell density was maintained between 3 × 10^5^ up to 3 × 10^6^ cells/mL by dilution of the cell suspension in the same growth medium.

### 2.3. Thermo-Resistant Cells

In this study, we generated thermo-resistant (TR) cells from CHO cells (named puro cells) previously developed in our laboratory [13]. Puro cells contain a single gene cassette in a specific chromosome location into which an interested protein gene can be readily inserted with recombination by replacing the *puro* gene. In order to obtain TR cells, puro cells were seeded in a 10 cm dish and grown in a CO_2_ incubator at 41 °C to 40–90% confluence; the harvested cells were diluted, reseeded, and grown at 41 °C again. This process continued till the growth rate became constant. As control, the cells at 37 °C grew to ~90% confluence; the cells were harvested, diluted, and reseeded again; the cells cultured at 41 °C were harvested and reseeded at the same time as those cells cultured at 37 °C, although they did not reach 90% confluence. In order to judge when the cells cultured at 41 °C became thermo-tolerant, the dynamics of cell number doubling time for the cells were monitored continuously. To this end, 3 × 10^5^ cells from those cultured at 41 °C were seeded in a 10 cm dish at the beginning of every week, grown at 41 °C for 5 days, and then the cell number was counted. During the same period, the same procedure was used to acquire the doubling times for the cells cultured at 37 °C. We use the following formula to calculate doubling time: N(t) = C2^t/d^ (N(t), where N is the number of cells at time t, d is doubling time (time taken to double cell number), and C is initial number of cells).

### 2.4. Cell Survival Assays

The cells cultured at 37 and 41 °C were harvested and diluted to 2 × 10^3^ cells/mL. One mL of cells were incubated at different temperatures for half an hour (1/2 h), separately. After that, the volume of 100 μL cells (about 200 cells) was seeded in a 10 cm dish, and the cells were incubated for 8 days at 37 °C before Giemsa staining and colony counting. Survival percentage was calculated as 100% × [(number of colonies at designated temperatures above 4 °C in dishes/number of colonies incubated at 4 °C in dishes)].

### 2.5. Transfection and Antibody Affinity Maturation

Generally, to replace the antibody gene integrated in the chromosome of CHO cells, CHO-puro cells were seeded into each well of a six-well plate. Cells were transfected with a mixture of 0.5 μg exchange vector, 2 μg pCI-Flp-2A-Cre, and 5 μL Lipofectamine^TM^ 2000 (Invitrogen, Waltham, MA, USA) for 6 h. After transfection, the cells were transferred into a 10 cm dish containing IMDM medium with 10% FCS. Afterward, the cells were collected and incubated with PE-conjugated anti-HA antibody (Abcam, Cambridge, UK, 1:250 in cold opti-MEM medium [Invitrogen, Waltham, MA, USA]) for 30 min at 4 °C, washed with cold opti-MEM once, resuspended in cold opti-MEM, and sorted for cells that expressed high displaying antibodies using FACSAriaIII (BD) flow cytometers.

To mature antibody affinity, CHO cells that displayed anti-TNFα antibodies were seeded into a six-well plate. The cells were transfected with 2 μg of pCEP4-Neo-AID and 5 μL of Lipofectamine^TM^ 2000 for 6 h, washed, and maintained in IMDM containing 10% FCS and HT for one day; then, the cells were expanded in IMDM with 10% FCS, HT, 1 mg/mL neomycin for 12 days and incubated with PE-conjugated anti-HA antibody and TNFα-GFP (PE-conjugated anti-HA antibody: Abcam, 1:250; TNFα-GFP: 1:10,000 in cold opti-MEM medium [Invitrogen, Waltham, MA, USA]) for 30 min at 4 °C or 42 °C, and sorted for cells that expressed high-affinity antibodies using FACSAriaIII (BD) flow cytometers. Generally, 1 μg of TNFα-GFP antigen and 8 μg of anti-HA antibody were used to label 2 × 10^7^ cells.

### 2.6. Detection of Antibody Mutations

The genomic DNA of sorted cells displaying antibodies was purified with a genomic DNA purification system (Promega, Madison, WI, USA). PCR with the following primers produces target gene fragments containing the same endonuclease sites with the plasmid vector at 5′ and 3′ ends. After enzyme digestion, it can be inserted into pMD-19 T-vector. Primers used for antibody gene PCR were as follows:CMV-F: CGCAAATGGGCGGTAGGCGTG;TM-R: CTGCGTGTCCTGGCCCACAGC.

The clones were sequenced, and the mutations were identified by comparing the sequences of these clones with the sequence of the wild-type antibody gene.

### 2.7. Protein Expression and Purification

Antibodies were prepared by following the manufacturer’s protocol (Invitrogen, Waltham, MA, USA). Briefly, anti-TNFα wild-type (WT) and mutant (G94R) scFvs were transfected into 293F cells (Invitrogen, Waltham, MA, USA) and cultured for 6 days. Then the supernatants were collected, anti-TNFα scFvs were affinity-purified using nickel columns (Sangon Biotech, Shanghai, China).

### 2.8. Antibody Thermal Stability Measurements

#### 2.8.1. Differential Scanning Calorimetry (DSC)

Thermal unfolding profiles of anti-TNFα antibody variants were measured by DSC using the Nano DSC system (TA Company, New Castle, DE, USA). All antibodies were tested in PBS, pH 7.4 in the protein concentration range of 0.8–1.0 mg/mL at a scan rate of 1 °C/min. Samples were heated from 30 to 100 °C.

#### 2.8.2. Thermofluor Assay

Thermofluor assay analysis was carried out using the Quantstudio 7 Flex (Applied biosystems, Waltham, MA, USA). As the temperature increases, upon reaching the dissolution temperature (Tm), the normal protein conformation is destroyed, the hydrophobic region is opened up, the dye (syproR orange protein dye) binds to the hydrophobic region, and then fluorescence emits and is recorded. The protein concentration was 20–40 μM, and the volume was 24 μL. The dye was added to the protein solution to the final 25× concentration as suggested by the manufacturer. The well-mixed samples were heated from 25 to 95 °C at 1 °C/min to detect Tm (dissolution temperature).

#### 2.8.3. Antibody Affinity Measurement

The affinities of various antibodies were measured by the Octet biomolecular interaction technology (ForteBio Octet, Menlo Park, CA, USA). TNFα-GFP protein were conjugated with biotin according to the manufacturer (ForteBio Octet) and concentrated at 20 μg/mL. The concentrated antigen was added and fixed to Streptavidin (SA) biosensors. The preparation of the antibody proteins was described above. The detection conditions used were (1) baseline 240 s; (2) loading 240 s; (3) baseline 180 s; (4) association 120 s with a series of concentrations (8000 nM, 4000 nM, 2000 nM, 1000 nM, 500 nM, 250 nM for wild-type antibody; 1000 nM, 500 nM, 250 nM, 125 nM, 62.5 nM, 31.25 nM for mutant antibody) of αTNF-α scFv; (5) dissociation 180 s. The K_on_ and K_off_ rates were measured by Octet software, and K_D_ was calculated for each antibody mutation by the K_off_/K_on_ ratio.

### 2.9. Analytical Ultracentrifugation

Sedimentation equilibrium experiments were performed on a Beckman Optima XL-I analytical ultracentrifuge as described [14]. Briefly, each antibody protein sample (400 μL, 0.5 mg/mL) was added to a sample slot, and the machine was run at 5000 g for 6 h, and the distributions of mono and aggregated proteins were analyzed.

### 2.10. Maturation of Antibody for Improved Stability and Affinity

A protocol developed by Chen et al. [13] was used with minor modifications for antibody evolution. Briefly, the obtained TR cells bearing anti-TNFα scFv gene were transfected with an activation-induced cytidine deaminase (AID) expression vector for mutagenesis. After culturing for 12 days at 37 °C, the TR cells were cultured at 41 °C for one more day before being labeled with GFP-TNFα and anti-HA antibody (Abcam, 1:250 in cold opti-MEM medium [Invitrogen, Waltham, MA, USA]) and flow cytometric sorting for the cells highly displaying the antibodies with the highest antigen-binding abilities. FACS AriaIII (BD) and FACS Calibur flow cytometers were used for cell sorting and analysis, respectively. Then, the collected cells were proliferated for about 20 days, and the next round of sorting was performed (in about 12 days). Fifty scFvs mutant genes were cloned from the cells after one and two rounds of evolution, and sequenced for the information of point mutation distributions in about 5 days. Thus, the whole process is 50 days. The procedure is illustrated in Figure 2.

### 2.11. Prediction of the 3D Structure of the Mutation

In order to analyze the mechanism underlying the improved stability resulted from the mutation G94R (refer to Figures 5 and 9), the antibodies were numbered using Chothia scheme by AbRSA (http://cao.labshare.cn/AbRSA/, accessed on 15 June 2019). The 3D structure was predicted using ABodyBuilder [15]. The mutant was further modeled using CISRR [16]; thus, the conformation changes could be observed by Pymol.

## 3. Results

### 3.1. Generating Thermo-Resistant Cells

A key factor limiting the increase in antibody stability using mammalian cell-based display is that mammalian cells are not as resistant to high temperatures as yeast or bacterial cells and often die after heat shock, making it difficult to continue the next round of evolution. Therefore, the development of thermo-resistant (TR) mammalian cells will help the evolution of antibody stability. To this end, we cultured CHO cells that contained a gene cassette for the integration of a gene of interest in an incubator at 41 °C and monitored their growth state continuously. The reason that we chose 41 °C is that long-term thermo-tolerance of CHO cells can be developed at and below 42 °C but not above 42 °C [17]. The choice of 41 °C as the culturing temperature is also due to the ±0.5 °C error range of a regular thermometer.

At 37 °C, the time for doubling cell numbers in a culture dish was about 21.5 h. The doubling time was sharply increased when the temperature was shifted from 37 °C to 41 °C, then attenuated after the first week at 41 °C, and reached a plateau of around 25 h after 5 weeks of culture at 41 °C (Figure 1). We noticed that many cells floated in culture dishes during the first week after shifting from 37 °C to 41 °C, suggesting that the sharp increase in the doubling time is partly because a lot of cells died. After cells were incubated at 41 °C for 5 weeks or longer, there were only a few floating cells in dishes compared to those cultured at 37 °C, suggesting that the higher culturing temperature slightly prolongs the time needed for cell proliferation instead of killing cells.

The evolution procedure for more stable antibodies includes the following two steps: (1) generating antibody mutants displayed on cells and (2) revealing the differences in heat resistance of these mutants by incubating cells at 41 °C (Figure 2). AID enzyme is sensitive to heat; thus, mutating the antibody was carried out by culturing cells at 37 °C for about 12 days after transfection of the AID gene into cells. Afterward, cells were cultured at 41 °C for 24 h before labeling and flow cytometric sorting to enrich cells displaying thermo-resistant and high-affinity antibodies. For this purpose, the cells have to be not only thermo-resistant but also persistent in thermo-resistance. We tested the persistence of thermo-resistance of the cells developed by continuously culturing at 41 °C for 7 weeks. We found that either after growing the cells at 37 °C for 12 days or storing them in liquid nitrogen, and thawing the cells, or after inserting an antibody gene into the gene cassette, the cells were still thermo-resistant (Appendix A). Therefore, the cells are suitable for the evolution procedure.

### 3.2. Conditions for Labeling Cells

In order to have a sufficiently high viable rate of sorted cells, we usually choose 4 °C, at which we label cells with reagents to monitor antibody display and the antigen-binding ability for 1/2 h. In this study, to reveal the thermo-resistant differences of the mutated antibodies displayed on CHO cells, we had to label the cells at a higher temperature. We examined the effects of 1/2 h incubation at various temperatures on survival rates of control (ctl) and TR CHO cells and found that the survival rates of both cells reduced significantly when incubated at 37 °C or above (Figure 3a). In particular, at 42 °C or 43 °C at which we aimed to label cells, the majority of cells died. These survival rates are not suitable for an efficient antibody evolution. In the labeling process, we want to minimize the loss of cells so as to ensure the diversity of the mutation library. BSA (Bovine Serum Albumin) is usually used as a stabilizer to protect protein activity under adverse conditions such as heating. It is also often used as a protein supplement in cell culture media. We speculate that the addition of BSA may make the cell membrane and intracellular proteins more stable, thus improving the survival rate of cells. We performed similar experiments again by adding BSA (5% of the final concentration) to the labeling buffer and found that BSA greatly enhanced the survival rates (Figure 3b). The survival rate of TR cells at 42 °C reached 80% and met our requirement for antibody evolution.

We were curious how incubation at 42 °C for 1/2 h influences cell labeling. We labeled TR cells displaying anti-TNFα antibodies with anti-HA antibodies (for detecting antibody display) and GFP-TNFα (for monitoring antibody affinity). The displays and antigen bindings at 4 °C and 42 °C were almost the same (Figure 4). Antigen-binding was much reduced in the samples without BSA compared to the samples with BSA at either temperature. Based on the above results, we labeled cells at 42 °C for 1/2 h at the presence of 5% BSA for antibody maturation described in the following experiments.

### 3.3. Evolution to Simultaneously Improve Antibody’s Stability and Affinity

Antibody stability improvement was carried out with two rounds of cell proliferation-flow cytometric sorting. For the first round of evolution, antibody-displayed TR cells were transfected with AID-expressing plasmid and proliferated (antibody gene mutations were accumulated during this period) in a neomycin-containing medium (to keep the AID-expressing plasmid in cells) for 12 days, harvested and labeled with PE-conjugated anti-HA antibody and GFP-TNFα. The labeled cells with the highest GFP as well as PE fluorescence (the top 0.01%, around 6000 cells) were collected by flow sorting out of roughly 8 × 10^7^ cells (Figure 5a). After the collected cells were grown for 20 days, the cells (about 8 × 10^7^ cells) were subjected to the second round of sorting, and the collected cells (about 6000 cells) grew to more than two million for antibody gene cloning (in about 12 days). Thirty-nine clones were sequenced. Twenty clones had only a single identical point mutation on the 94th amino acid residue (from glycine to arginine, a change G94R at base level). The mutated residue is located right before the heavy chain CDR3 region (Appendix A). The other 19 clones contained no mutations. In comparison, the antibody maturation was performed under normal conditions in which cells were cultured at 37 °C and labeled at 4 °C. The frequency and diversity of mutation obtained from the first round of evolution were much higher than those acquired at higher temperatures (Appendix A). After two rounds of evolution, although frequencies of mutation under both maturation conditions reached high levels, the diversity of mutation under normal temperature conditions (5 different variants) was much higher than that obtained under high-temperature conditions (only one variant). These results suggest that incubating and labeling at higher temperatures is more challenging for newly generated mutants to survive.

We compared the most enriched clone A115V obtained under normal conditions and the only clone G94R under high-temperature conditions. We separately displayed G94R, A115V, and WT antibodies on CHO cells and analyzed them with flow cytometry. Being incubated at 4 °C, the antigen-binding abilities of both mutant antibodies were obviously higher than that of the WT antibody. Labeling at 42 °C, the binding ability of G94R was higher than both A115V and WT antibodies (Figure 5b). The difference in the antigen-binding abilities of the G94R labeled at 4 °C and 42 °C was not much different. In contrast, the labeling at 42 °C for 30 min reduced the binding abilities of the WT and A115V antibodies to significantly lower levels. The results suggest that the G94R has a higher affinity than the wild-type and that the increased affinity to the G94R is largely independent of the labeling temperatures.

To confirm the increased stability and binding ability of the G94R antibody, we synthesized the WT and mutant antibodies in 293F cells. The total cell culture volume for each of the two antibodies was 120 mL and yielded 2.2 mg WT and 7.5 mg G94R scFvs, hinting that the thermo-stability might be an important factor in guaranteeing high-level expression of this antibody.

We examined the thermal stability of the antibodies with differential scanning calorimetry (DSC). The dissolution temperature (Tm) of the WT and mutant antibodies were 50.20 °C and 61.98 °C, respectively (Figure 6). We also performed thermofluor assay on the two antibodies, and the results verified those from the DSC assay (Figure 7).

We compared WT and G94R antibodies for their dissociation constant (K_D_) using the Octet biomolecular interaction technology. The K_D_ values of the WT and mutant antibodies were 1.55 μM and 9.69 nM, respectively (Table 1). The affinity of the mutant antibody was 160-fold higher than that of the wild-type, and both a slower dissociation rate of G94R scFv-antigen complex and a faster rate of the association of G94R scFv with the antigen (TNFα) contributed to the increase on affinity (Table 1, Appendix A).

Obviously, the thermal stability and the affinity of the antibodies were both improved under high-temperature conditions. We wondered whether the enhanced stability was a major element of the affinity improvement for the G94R antibody. To this end, we monitored the molecular masses of the WT and G94R antibodies with sedimentation velocity analytical ultracentrifugation. The theoretical molecular weight of monomeric WT and G94R scFvs with a 6-linked His tag are approximately 25 kDa, and the sedimentation velocity analysis demonstrated that the molecular weights of the main components of the mutant and WT antibodies were very close to each other at 24 kDa (Figure 8 and Appendix A). Both of them had small amounts of dimers and degraded fragments, but not enough to impact their affinity (Appendix A). The results suggest that the increased stability is not a dominant factor behind the large improvement in the affinity of the G94R antibody.

The G94R mutation site is at position 94 of the heavy chain in the Chothia scheme, located right before the heavy chain CDR3 region. This indicates that this position could be critical for the stability of the CDR3 loop. The mutation of Gly to Arg at 94 would result in a favorable energy decrease, from −349 to −356.0 according to CISRR scoring. Through inspecting the 3D structures, the shortest distance between the guanidine group of the mutation Arg94 and the acidic side chain of Asp100A at the end of CDR3 is 2.8 Å, which would form a strong salt bridge that further stabilizes the protein structure (Figure 9, Appendix A). In addition, the two other residues following this mutation are both positively charged Arg residues, implying that they may interact with a negatively charged region of an epitope on TNFα, which could further enhance the electrostatic interaction between the antibody and antigen.

### 3.4. The Affinity and Stability Improvement of the Two Other Antibodies

To demonstrate that the above success in simultaneously improving affinity and thermal stability is not unique to this specific anti-TNFα antibody, we also used this method to simultaneously improve the affinities and stabilities of two other antibodies (anti-CD19 scFv and anti-human kappa chain of CL). After two rounds of cell proliferation-flow cytometric sorting, 50 clones were sequenced for each of the two antibody strains. Then we choose the enriched clones to express their soluble scFv protein in 293F cells.

As for the anti-CD19 antibody, three mutants were enriched, which were P155S, P155A, and P155T. All three mutations occurred on the same amino acid residue, P155. We determined the affinities and dissolution temperature of the wild-type and mutants and found that both the affinities and thermal stabilities of the three mutants were higher than WT (Figure 10 and Figure 11). The K_D_ values of the WT, P155S, P155A and P155T were 0.95 nM, 0.29 nM, 0.19 nM and 0.068 nM, respectively. The dissolution temperature (TM) of the WT, P155S, P155A, and P155T were respectively 59.32 °C, 62.43 °C, 62.73 °C, and 62.36 °C (Table 2). Interestingly, the three mutants have similar higher Tm values while the affinities, though significantly improved compared to WT, are quite different. We were curious what might be the mechanisms behind the improved thermostabilities and affinities of the P155S, P155A, and P155T antibodies. We first employed a well-established antibody 3D structure modeling tool, ABodyBuilder [15], to construct the WT, P155S, P155A, and P155T structures. We refined the predictions by energy minimization using NAMD. The 3D structures implied that the protein stability alternations produced by mutations are due to the changes in free energies (PremPLI). The results showed that all three substitutions of Pro with Ala, Ser, or Thr decreased the free energy, which indicates that they are favorable for antibody stability (Table 3). We then compared the conformation changes upon the single residue substitution at position 30 of the heavy chain. We observed that the overall structures are almost the same except for some significant alternations on the loops near position 30. Surrounding the Pro30, loops including CDR2 showed significant shifts upon substitution P155S, P155A, or P155T (Figure 12). It is attributed to the backbone preference of Pro other than Ala, Ser, and Thr. Pro does not contain the amino group -NH_2_ but is rather a secondary amine. The Pro side chain forms a pyrrolidine loop, which changes the protein backbone preference. In contrast, Ala, Ser, and Thr are more flexible, and the substitutions relaxed the loop structure, making it more stable, and it is also the possible reason that the substitutions relaxed the CDRs binding with the antigen resulting in stronger binding affinities.

As for the anti-human kappa chain of CL, the five mutants S20N, S20N-A169T, S29N, T24N-K46N, and S86N-A169S were enriched. We synthesized the WT and these mutants in 293F cells. The cell culture volumes were 100 mL for either WT and all the mutants and yielded 3 mg WT and 12 mg mutant clones, respectively. The yields of these five mutants were obviously much higher than that of WT, but only one mutant (S86N-A169S) has a higher affinity and Tm (dissolution temperature) (Figure 13 and Figure 14). The K_D_ values of the WT and S86N-A169S were respectively 15.5 nM and 5.9 nM. The Tms of the WT and S86N-A169S were respectively 65.25 °C and 70.02 °C (Table 4).

## 4. Discussion

In this study, we developed a procedure performing CHO cell display at temperatures above 41 °C to simultaneously mature antibody affinity and stability. Two rounds of evolution achieved the affinity and thermal stability improvement on all three antibodies.

To our knowledge, this is the first study combining mammalian cell display and in vitro evolution to enhance antibody stability. Mammalian cells are advantageous in protein folding, codon usage, and post-translational modifications for mammalian proteins [18]. CHO cells have become the most used in producing therapeutic antibodies and other medicinal proteins (Schmidt 2004; Walsh and Jefferis 2006). Antibodies and proteins derived through evolution using CHO cell display should be easily mass-produced using a CHO cell as a host. Human proteins and antibody constant fragment (Fc)-conjugated proteins or protein domains are frequently used for the treatment of various diseases, e.g., recurrent pericarditis, ankylosing spondylitis, and psoriasis [19,20,21]. Many human proteins are posttranslationally modified and potentially useful in treating diseases, and the procedure described here could help to develop some of these proteins into drugs. This system has the advantage of improving the stability of the proteins that need post-translational modifications for their functions. For example, we succeeded in generating a programmed death 1 (PD1, a glycoprotein) mutant via directed evolution with CHO cell display, which is much more stable and has an affinity for PD-1 ligand (PD-L1) of over 100-fold higher than the wild-type PD1 (our unpublished data), and the mutant PD1-Fc has an opportunity to be developed into a cancer drug due to its advantages regarding high affinity to PD-L1. 

Yeast and bacteria are more resistant to heat shock, and displays using these microbes are readily used to generate mutant antibodies with improved stability at temperatures above 37 °C. In this case, yeast cells were incubated at 79 °C for 10 min before labeling at 4 °C. All yeast cells died in this study, but the investigators were able to clone mutant antibody genes and successfully obtained a much more stable antibody [22]. We also tested labeling CHO cells at 45 °C for 1/2 h, and many thermo-resistant cells were broken (data not shown). Therefore, we chose 42 °C for labeling CHO cells displaying antibody mutants and for maturing antibodies. Our sorting criteria also included the highest levels of displayed antibodies apart from the highest antigen-binding abilities, and these criteria served our purpose well.

We obtained an anti-TNFα mutant antibody with a Tm (dissolution temperature) value of 12 °C higher and affinity 160-fold greater than its parent antibody using our evolution procedure (Figure 6 and Figure 7, Table 1). Only one mutant antibody was identified after two rounds of the anti-TNFa scFv evolution. We sequenced 71 antibody gene clones isolated from the cells after the first round of evolution, but there were two identical mutants (C156S) and two different frame-shifted mutants. G94R mutant that was isolated from the cells after the second round of evolution and conferred improved stability and affinity did not exist among the 71 antibody gene clones after the first round of evolution, suggesting that G94R is rare after the first round of maturation or induced by AID enzyme during the second round of evolution.

Modeling the anti-TNFα scFv before and after the mutation suggests that the G94R increases the stability of the heavy chain CDR3 by forming a salt bridge between Arg 94 and Asp 100A. This change might also enhance the affinity by increasing electrostatic interaction with the antigen. A definite conclusion needs further structural examinations, such as crystallographic analyses.

We notice that few mutant antibodies were generated from the first round (4 mutations and 3 types/71 clones) and the second round (19 mutations and 1 type/39 clones) of the evolution of the anti-TNFα antibody. We also performed one round of evolution by culturing at 37 °C and labeling at 4 °C and obtained 11 points of mutations and three types of mutation (Appendix A). None of the three types of mutation is the same as those identified from the evolution at a temperature above 37 °C. It is likely that culturing cells at 41 °C for 24 h before labeling and labeling cells at 42 °C discarded these mutants. This phenomenon is probably unique to this antibody because we were able to obtain multiple antibody mutants with simultaneously improved affinities and thermal stabilities from the evolution of the other two antibodies (refer to the following statements).

Three anti-CD19 scFv mutants were obtained through evolution. However, when we examined the anti-human kappa chain of CL, the affinity with flow cytometry of these five mutants was higher than that of WT, but only one mutant had a higher affinity and Tm. We suspected that not all mutants with high affinity measured by flow cytometry are good mutants. Nevertheless, the procedure is likely to be widely used in maturing antibodies and other proteins in the future.

Usually, 5–12 rounds [13,23] of cell proliferation-flow sorting are carried out to mature antibodies against antigen or receptor against its ligand for higher affinities [24]. We published an article aimed only at achieving affinity maturation of antibodies using CHO cell display [25]. In addition, antibody engineering with a mammalian display by other research groups includes Robertson’s and See’s works [26,27]. Here, we just performed two rounds of evolution and significantly improved both stability and affinity (Figure 6 and Figure 7, Table 1). Therefore, the current procedure is much more efficient than those described previously in the literature. We expect that this efficient procedure may find a wide range of applications in maturing antibodies and proteins for the improvement of both affinity and stability.

In this study, we improved both affinity and thermo-stability of three different scFvs. As reported, the affinity enhancement of an scFv cannot guarantee the affinity improvement of its corresponding full-length antibody [28,29,30]. Previously, we reported success in improving the affinity of full-length antibodies using the CHO cell display at normal temperature (37 °C) [31]. It is worthwhile to mature full-length antibodies using the procedure described in this study to find out its efficiency for simultaneously improving the affinity and stability of full-length antibodies.

In conclusion, we successfully established a thermos-resistant CHO cell line for simultaneous maturation of thermodynamic stability and affinity of antibodies. This is the first report on thermodynamic stability in mammalian cells, and this procedure is likely to be widely used in maturing other proteins which require post-translational modification in the future. 

## Figures and Tables

**Figure 1 bioengineering-09-00360-f001:**
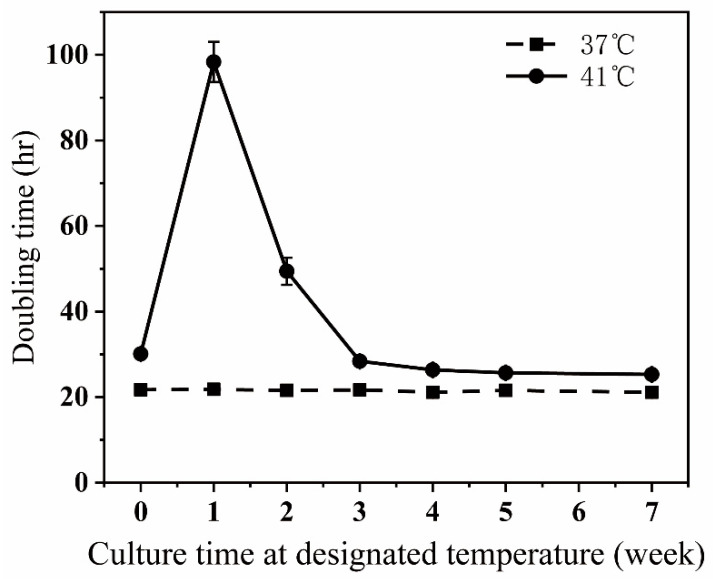
The doubling times of the cells incubated at 37 °C and at 41 °C. Cells were incubated at either 37 °C or at 41 °C continuously for 7 weeks. At the beginning of every week, a portion of cells were removed from the main cultures and seeded for the doubling time measurement. Refer to Materials and methods for the experiment procedure and calculation of the doubling times. The results were derived from three independent repetitive experiments.

**Figure 2 bioengineering-09-00360-f002:**
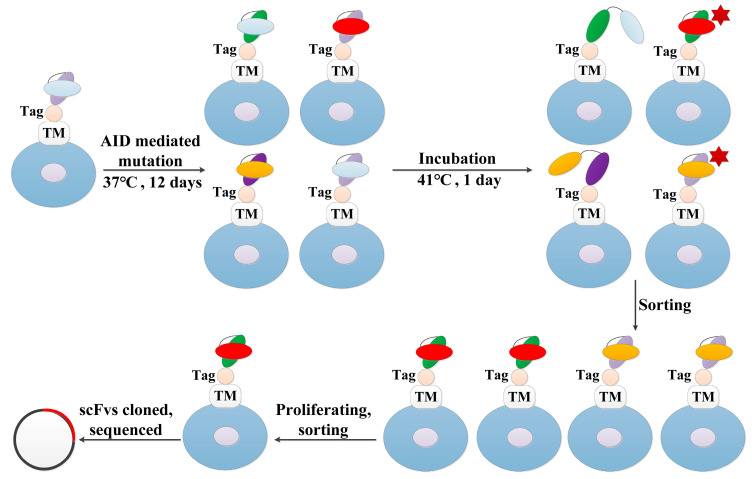
Procedure selecting more stable antibody variants. The TR cells bearing anti-TNFα scFv gene were transfected with AID gene for antibody mutagenesis. After culturing for 12 days at 37 °C to generate mutant scFvs (two linked-ovals), the cells were cultured at 41 °C for 24 h before being labeled with GFP-TNFα star symbol and anti-HA antibody at 42 °C, and flow cytometric sorting for the cells highly displaying the antibodies with the highest antigen-binding abilities. Then, the collected cells were proliferated, and the next round of sorting was performed. Tag: Human influenza hemagglutinin for monitoring antibody display level. TM: transmembrane peptide for anchoring scFv to the surface of the cell.

**Figure 3 bioengineering-09-00360-f003:**
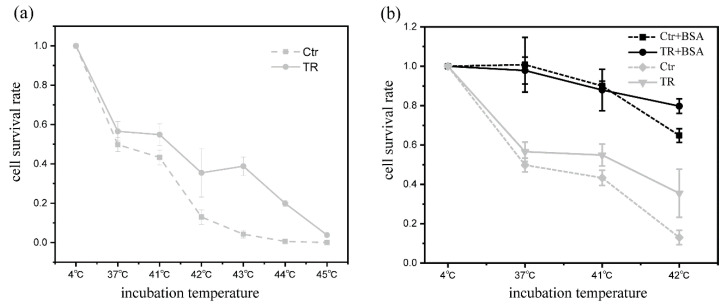
Effects of the incubation temperature and BSA on cell survival. (**a**) The survivals in medium without BSA. (**b**) The survivals in medium containing 5% BSA. The ctl (the cells cultured at 37 °C) and TR (the cells cultured at 41 °C) cells were incubated at the designated temperatures for 1/2 h, separately. Afterwards, the cells were grown at 37 °C, the formed colonies counted, and the survival rates calculated by following the procedure in Materials and methods. The results were derived from three independent repetitive experiments.

**Figure 4 bioengineering-09-00360-f004:**
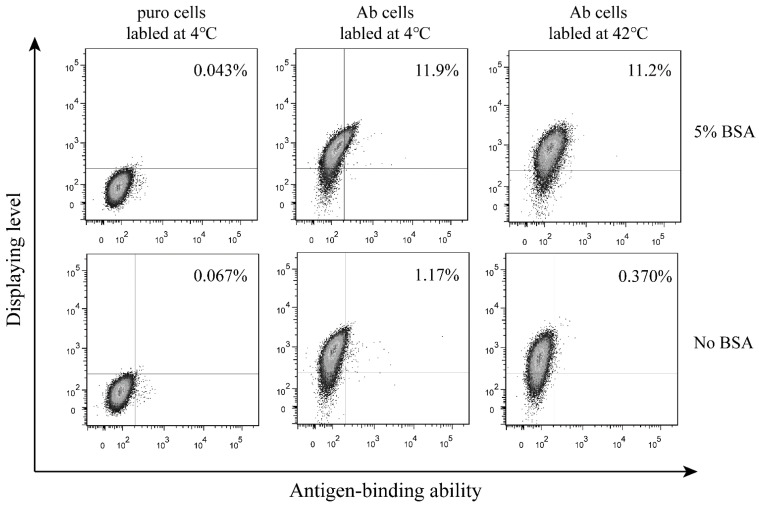
Effects of different labeling conditions on antibody display and antigen-binding detected with flow cytometry. TR cells displaying anti-TNFα antibody (Ab) were labeled for detecting antibody display and for monitoring antibody affinity at 4 °C and 42 °C for 1/2 h with and without BSA (5%), then subjected to flow cytometric analysis.

**Figure 5 bioengineering-09-00360-f005:**
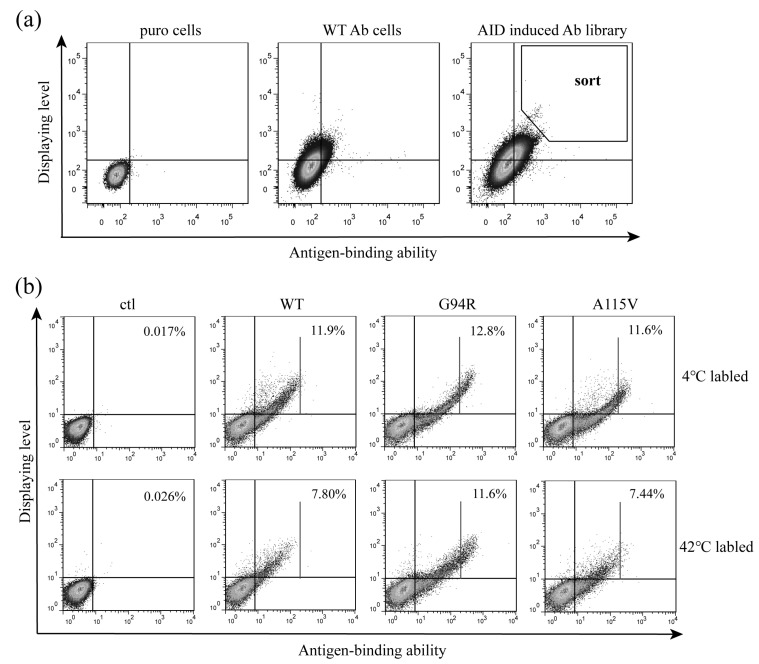
Stability and affinity maturation of anti-TNFα antibody. (**a**) Antibody maturation using TR cells. The maturation was conducted by following the procedure demonstrated in Figure 2. The sorting gate was set to collect the cells with the highest antigen binding and the highest displaying level ability (the top 0.01%). Two rounds of sorting were carried out in the same manner, but only the first round of sorting is shown here. (**b**) Analyses of the display levels and antigen-binding abilities of the wild-type (WT) and mutant (G94R and A115V) antibodies labeled at 4 °C and 42 °C. The assays were made on cells transiently transfected with the WT, A115V and G94R antibody genes, respectively, and then were labeled at 4 °C and 42 °C before flow cytometric analysis.

**Figure 6 bioengineering-09-00360-f006:**
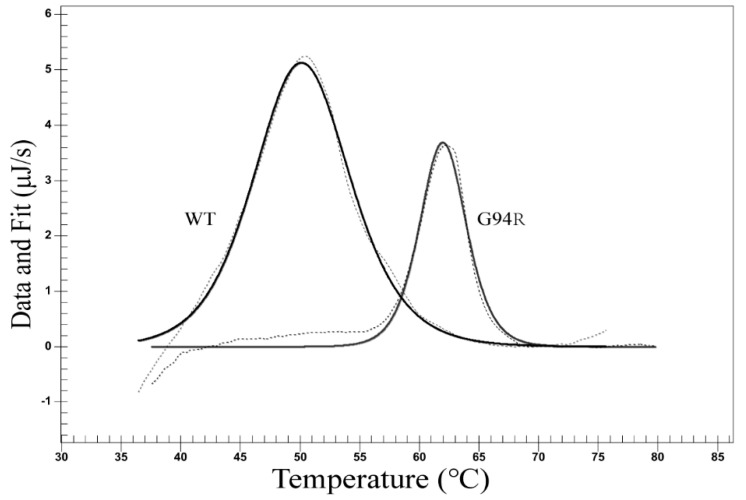
Differential scanning calorimetry (DSC) analysis on antibody stability. The dissolution temperature (Tm) of the WT and mutant scFvs (G94R) were 50.20 °C and 61.98 °C, respectively. The dashed line shows the raw data, and the solid line shows the fitting curve.

**Figure 7 bioengineering-09-00360-f007:**
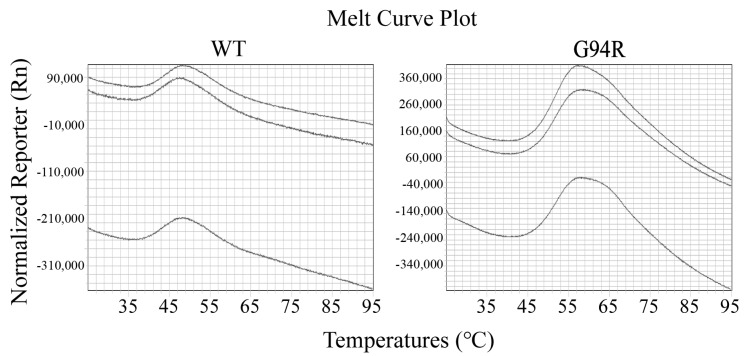
Thermofluor assay for the dissolution temperatures (Tm) of the wild-type and G94R antibodies. Thermal stability of the WT and mutant scFvs were monitored using SYPRO Orange staining and detected using the Quantstudio 7 Flex. As the temperature increases, upon reaching Tm, the protein conformation is destroyed, and the dye can bind to the hydrophobic region. Three repetitive tests were carried out, and only one result was demonstrated here.

**Figure 8 bioengineering-09-00360-f008:**
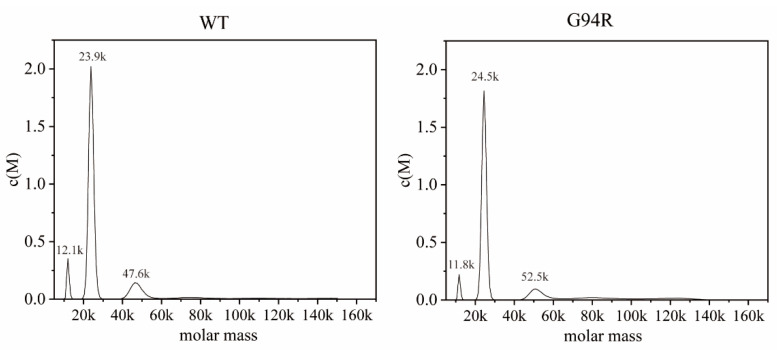
Sedimentation velocity analytical ultracentrifugation of WT and G94R proteins. The molecular weight of monomeric of WT and G94R with a 6 × His tag is approximately 24 kDa.

**Figure 9 bioengineering-09-00360-f009:**
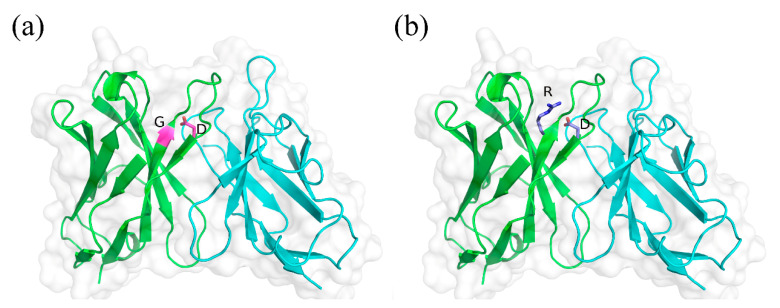
Structural modeling analysis on the mechanism of the improved stability of G94R antibody. Structures of the wild-type and G94R mutant were constructed and modeled by using ABodyBuilder and CISRR programs (Refer to Materials and methods). (**a**) Gly at 94 and Asp at 100 A are shown in purple sticks. (**b**) Arg at 94 an Asp at 100 A are shown in blue sticks. The salt-bridge formed between the two residues, which stabilizes the loop structure of CDR3.

**Figure 10 bioengineering-09-00360-f010:**
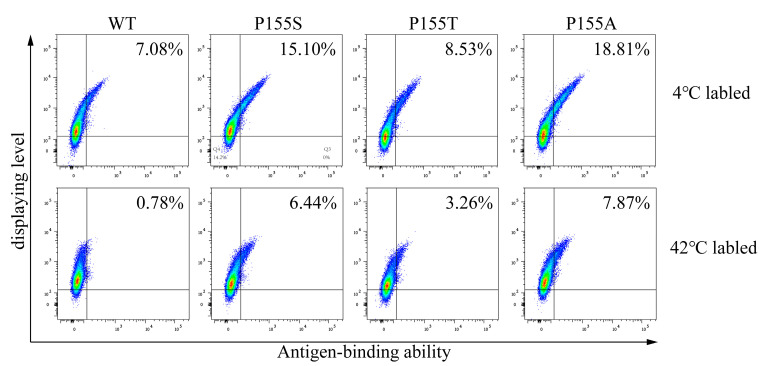
Analyses of the display levels and antigen-binding abilities of wild-type (WT) and mutants (P155S, P155A and P155T) labeled at 4 and 42 °C. The antibody genes were transiently transfected into cells, respectively, and then were labeled at 4 and 42 °C before flow cytometric analysis.

**Figure 11 bioengineering-09-00360-f011:**
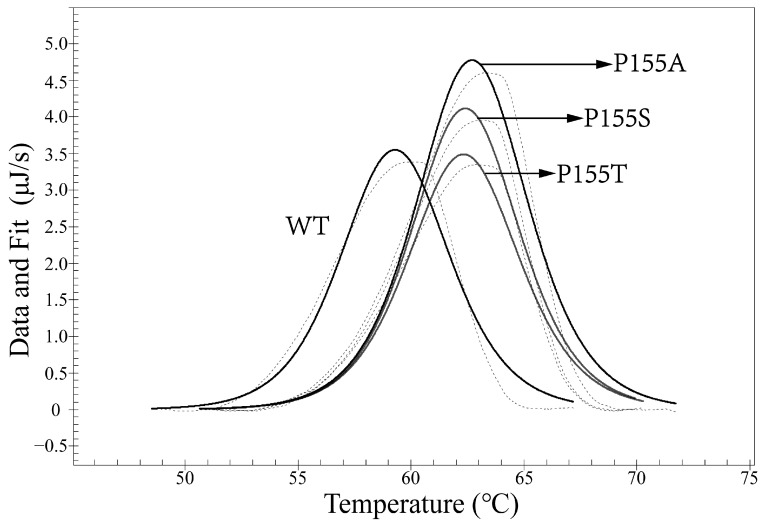
Differential scanning calorimetry (DSC) analysis on antibody stability. The dissolution temperature (Tm) of the WT, P155S, P155A, P155T were 59.32 °C, 62.43 °C, 62.73 °C and 62.36 °C, respectively. The dashed line shows the raw data, and the solid line shows the fitting curve.

**Figure 12 bioengineering-09-00360-f012:**
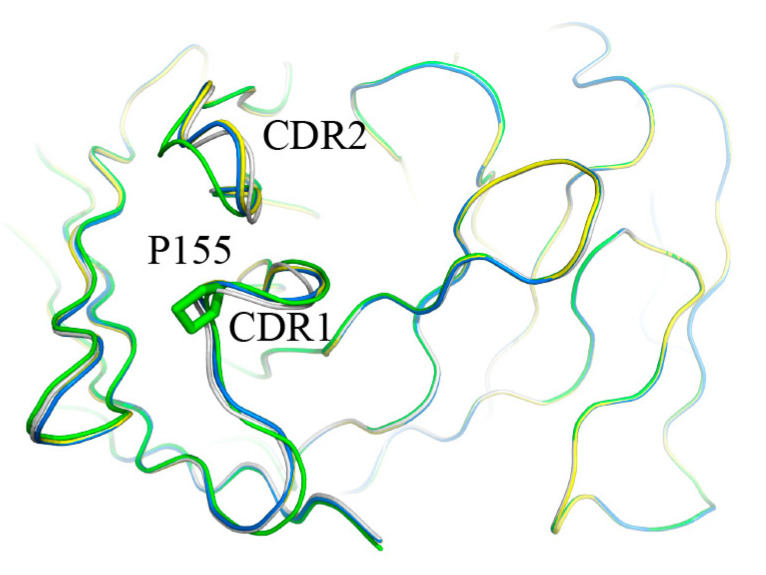
The modelling 3D structures of the antibodies. Green: wild-type antibody. Blue: P155A. Yellow: P155S. Gray: P155T. Pro at position 30 of heavy chain are highlighted by stick model. The nearby loops including CDR2 showed conformation changes induced by the residue substitutions at position 30.

**Figure 13 bioengineering-09-00360-f013:**
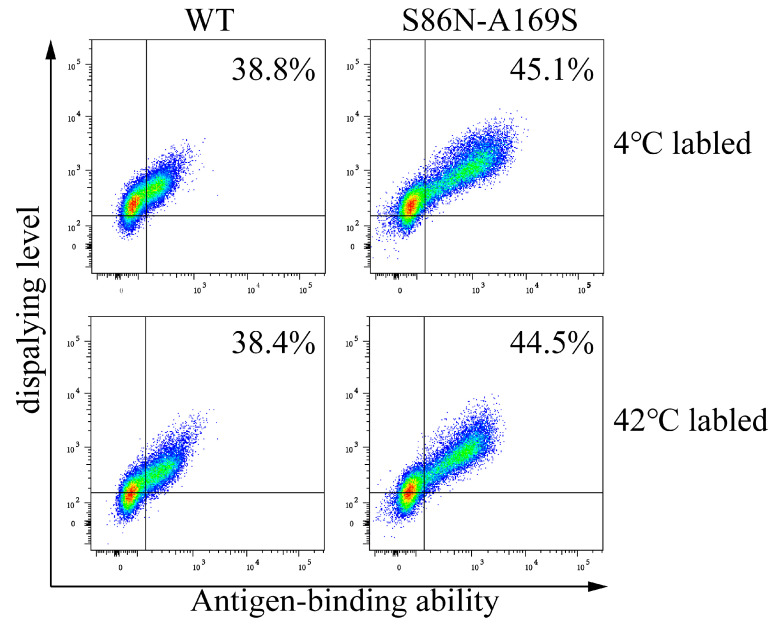
Analyses of the display levels and antigen-binding abilities of wild type (WT) and mutants (S86N-A169S) labeled at 4 and 42 °C. The antibody genes were transiently transfected into cells, respectively, and then were labeled at 4 °C and 42 °C before flow cytometric analysis.

**Figure 14 bioengineering-09-00360-f014:**
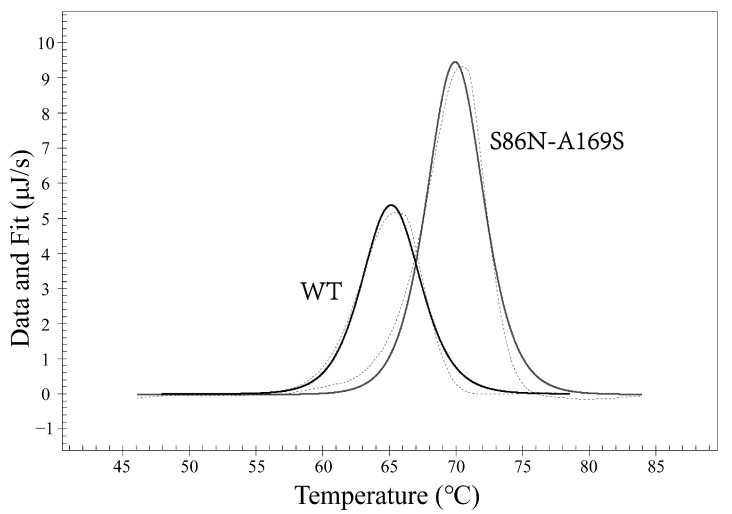
Differential scanning calorimetry (DSC) analysis on antibody stability. The melting dissolution temperature (Tm) of the WT and S86N-A169S were 65.25 °C and 70.02 °C, respectively. The dashed line shows the raw data, and the solid line shows the fitting curve.

**Table 1 bioengineering-09-00360-t001:** Affinities of different mutations of anti TNFa scFv.

Clone (Base)	Clone (Amino Acid)	K_D_ (M)	K_off_ (1/s)	K_on_ (1/MS)
WT	No	1.553 × 10^−6^	8.727 × 10^−3^	5.618 × 10^3^
G316A	G94R	9.688 × 10^−9^	1.586 × 10^−3^	1.637 × 10^5^

The values of K_on_ represented the association rates, the values of K_off_ represented the dissociation rates, and K_D_ = K_off_/K_on_.

**Table 2 bioengineering-09-00360-t002:** Affinities of different mutations of CD19 antibodies.

Clone (Base)	Clone (Amino Acid)	K_D_ (M)	K_off_ (1/s)	K_on_ (1/MS)
WT	WT	9.499 × 10^−10^	7.120 × 10^−4^	7.496 × 10^5^
C463T	P155S	2.857 × 10^−10^	1.860 × 10^−4^	6.512 × 10^5^
C463G	P155A	1.874 × 10^−10^	1.290 × 10^−4^	6.869 × 10^5^
C463A	P155T	6.769 × 10^−11^	5.680 × 10^−5^	8.383 × 10^5^

The values of K_on_ represented the association rates, the values of K_off_ represented the dissociation rates, and K_D_ = K_off_/K_on_.

**Table 3 bioengineering-09-00360-t003:** The predicted free energy changes upon residue substitutions at position 30 of heavy chain.

Mutation	ΔΔG
P155A	−0.19
P155S	−0.56
P155T	−0.71

**Table 4 bioengineering-09-00360-t004:** Affinities of different mutations of human kappa CL antibodies.

Clone (Base)	Clone (Amino Acid)	K_D_ (M)	K_off_ (1/s)	K_on_ (1/MS)
WT	WT	1.876 × 10^−8^	2.288 × 10^−3^	1.220 × 10^5^
G257A-G505T	S86N-A169S	5.287 × 10^−9^	7.352 × 10^−4^	1.391 × 10^5^

The values of K_on_ represented the association rates, the values of K_off_ represented the dissociation rates, and K_D_ = K_off_/K_on_.

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
