# Peer review of "Simultaneous Maturation of Single Chain Antibody Stability and Affinity by CHO Cell Display"

_bioengineering, 2022, doi:10.3390/bioengineering9080360_

Round 1

Reviewer 1 Report

1. Due to the importance of producing recombinant proteins and their stability, this article is an important work in the field.In addition, an increase in affinity has been done along with temperature resistancy.

2. The procedure, purpose and results are explained completely. (of course, the relevant check box was also sent in the refereeing format).

3. scfv is part of a complete antibody and differs in a number of features from complete antibodies, so it is recommended to mention scfv in the title of the article.

4. How long after the genetic manipulation of the mutation has been investigated?

5. What is the product of pcr method using primers? Further explanation of how it works?

Would you please write in method, centrifuge speed based on g(RCF). Is it also economical?

Author Response

Dear Reviewer,

We have tried hard to address all the questions and concerns raised by you on our manuscript titled “Simultaneous maturation of antibody stability and affinity by CHO cell display”. We have revised our manuscript extensively. The added and modified contents in the revised manuscript have been marked in red. I hope that our answers to the questions and the revised manuscript meet your requirements for its publication on bioengineering.

Sincerely yours,

Haiying Hang

Comments and Suggestions for Authors

  1. Due to the importance of producing recombinant proteins and their stability, this article is an important work in the field.In addition, an increase in affinity has been done along with temperature resistancy.

No answer or modification is needed.

  1. The procedure, purpose and results are explained completely. (of course, the relevant check box was also sent in the refereeing format).

No answer or modification is needed.

  1. scfv is part of a complete antibody and differs in a number of features from complete antibodies, so it is recommended to mention scfv in the title of the article.

As suggested, we have added single chain in the title of the article as following: Simultaneous maturation of single chain antibody stability and affinity by CHO cell display.

  1. How long after the genetic manipulation of the mutation has been investigated?

First, we used AID enzyme to generate antibody mutants at 37℃ for about 12 days, Afterwards, cells were cultured at 41℃ for 1 day to enrich cells displaying thermo-resistant and high affinity antibodies. After sorting, the collected cells were grown for 20 days, the cells were subjected to the second round of sorting, and the collected cells grew to more than 2 million for antibody gene cloning (in about 12 days). Finally, clones were sequenced in about 5 days. So, the whole process is 50 days. We have added the time line details in Materials and Methods.

  1. What is the product of pcr method using primers? Further explanation of how it works?

PCR process produces target gene fragments containing same endonuclease sites with plasmid vector at 5’ and 3’ ends. After enzyme digestion, it can be inserted into the plasmid. We have added this content in Materials and Methods.

Would you please write in method, centrifuge speed based on g(RCF). Is it also economical?

Thank you for your advice, we have modified them in the revised manuscript.

Reviewer 2 Report

Manuscript ID

bioengineering-1769815

Title

Simultaneous maturation of antibody stability and affinity by CHO cell display

Authors

Ruiqi Luo, Baole Qu, Lili An, Yun Zhao, Yang Cao, Shaohua Ma, Haiying Hang

This paper reported that the CHO cell display system for antibody selection. It seems to contain valuable information and well-organized. I think that it should be accepted after minor revision.

Comments

Well-organized and interesting paper.

Time course and antibody concentration.

The time course of cell concentration and antibody concentration are not clear. The authors should show or comment it in the revised manuscript

Author Response

Dear Reviewer,

We have tried hard to address all the questions and concerns raised by you on our manuscript titled “Simultaneous maturation of antibody stability and affinity by CHO cell display”. We have revised our manuscript extensively. The added and modified contents in the revised manuscript have been marked in red. I hope that our answers to the questions and the revised manuscript meet your requirements for its publication on bioengineering.

Sincerely yours,

Haiying Hang

Comments and Suggestions for Authors

Well-organized and interesting paper.

No answer or modification is needed.

Time course and antibody concentration.

The time course of cell concentration and antibody concentration are not clear. The authors should show or comment it in the revised manuscript

 We have added it in the revised manuscript as following: Generally, 1 mg of TNFa-GFP antigen and 8 mg of anti-HA antibody were used to label 2×107 cells.

Reviewer 3 Report

This is a nice study on adapting CHO cells to be thermo-resistant and using the obtained cell line for the isolate thermo-stable antibody scFv with improved affinity. To demonstrate the approach, antibodies of multiple targets have been tested. The work is well-designed overall with convincing data. I only have a few comments: 

1. scFv instead of Fab / IgG was the format used in this study on mammalian display / FACS. After scFv mutants have been identified, it will be ideal / more complete to convert the scFV clones to IgG and test again on affinity and stability, as it is not uncommon that format conversion (e.g. from svFB to IgG) may reduce the affinity. 

2. in addition to combinatory approaches (i.e. randomized libraries followed by selection), there are a few well established ways to improve protein / antibody stability by design, e.g. introducing disulfide bonds, salt bridges, rigid prolines etc. I wish authors could summarize a bit in introduction/ discussion on this topic. 

3. at the chosen temperature of 41C, with the usage of BSA, it seems there is no difference between Ctl CHO and TR CHO of their growth curves (Fig S3b). Was using the TR CHO necessary? 

4. Fig 11, what are the dashed and solid line curves? 

Author Response

Dear Reviewer,

We have tried hard to address all the questions and concerns raised by you on our manuscript titled “Simultaneous maturation of antibody stability and affinity by CHO cell display”. We have revised our manuscript extensively. The added and modified contents in the revised manuscript have been marked in red. I hope that our answers to the questions and the revised manuscript meet your requirements for its publication on bioengineering.

Sincerely yours,

Haiying Hang

Comments and Suggestions for Authors

This is a nice study on adapting CHO cells to be thermo-resistant and using the obtained cell line for the isolate thermo-stable antibody scFv with improved affinity. To demonstrate the approach, antibodies of multiple targets have been tested. The work is well-designed overall with convincing data. I only have a few comments:

  1. scFv instead of Fab / IgG was the format used in this study on mammalian display / FACS. After scFv mutants have been identified, it will be ideal / more complete to convert the scFV clones to IgG and test again on affinity and stability, as it is not uncommon that format conversion (e.g. from svFB to IgG) may reduce the affinity.

The reviewer is definitely correct on this point. As reported, the affinity enhancement of a scFv cannot guarantee the affinity improvement of its corresponding full length antibody (Holger et al. 2011; Liu et al. 2007; Steinwand et al. 2014). We used CHO display to mature both scFv and its corresponding full length antinbody, and found that some mutations generated through scFv were able to increase the affinity of its corresponding full length antibody and some were not (Wang et al. 2019). We have added the following in Discussion in the revised manuscript: "In this study, we improved both affinity and thermo-stability of three different scFvs. Previously, we reported the success in improving affinity of full length antibody using the CHO cell display at normal temperature (37°C). It is worthwhile to mature full length antibodies using the procedure described in this study to find out its efficiency for simultaneously improving the affinity and stability for full length antibodies in the future."

  1. in addition to combinatory approaches (i.e. randomized libraries followed by selection), there are a few well established ways to improve protein / antibody stability by design, e.g. introducing disulfide bonds, salt bridges, rigid prolines etc. I wish authors could summarize a bit in introduction/ discussion on this topic.  

We have added these approaches in the revised manuscript as following: "Stabilization engineering of antibodies can be accomplished in two completely different ways. One is through directed evolution and the other is through rational design of mutants for engineering modification.  The latter need to be based on certain structural analysis and rational design to obtain antibody mutants with high stability, and then after expression purification and stability test to verify whether it is thermal stability of the antibody. The two strategies may compensate for each other. The rational design is quick do to perform and may consider detailed intramolecular bonding mechanisms such forming new disulfide bonds. The evolution approach can reach out the thinking box, often resulting in stability with surprise ways."  

  1. at the chosen temperature of 41C, with the usage of BSA, it seems there is no difference between Ctl CHO and TR CHO of their growth curves (Fig S3b). Was using the TR CHO necessary?  

Since our labeling temperature is 42℃, not 41℃, with the usage of BSA, it can be seen from the figure that the survival rate of TR CHO cells is significantly higher than that of control CHO cells at 42℃, so it is very important to use TR CHO.

  1. Fig 11, what are the dashed and solid line curves?

In all scanning calorimetry (DSC) analysis figures(Fig 6, 11, 14), the dashed line shows the raw data, and the solid line shows the fitting curve. We have added the information in the revised manuscript as following:“The dashed line shows the raw data, and the solid line shows the fitting curve.”

Reviewer 4 Report

The research article by Luo et al. titled ‘Simultaneous Maturation of Antibody Stability and Affinity by CHO Cell Display' aims to present antibody maturation methodology to achieve both affinity and stability, utilizing CHO cell display. 

This field is a very popular field with multiple research papers published yearly. Although Luo and Hang’s current article is focused on optimization of both the interaction and stability of antibodies against antigens, the same authors published an article on this topic in May 2020 titled ‘High efficiency CHO cell display-based antibody maturation’ (Scientific Reports 2020, 10:8102). The aim of their previous research was to achieve affinity maturation of antibodies using CHO cell display. This topic is very similar to their current article and from the abstract, it can be deducted that the uniqueness in the current paper is the simultaneous maturation of antibody affinity as well as thermostability. Although the authors presented stability issues of the engineered antibodies, there is a lot of overlap of information between the two papers, both covering antibody maturation using AID as a diversity generator.

In addition, competing papers, regarding antibody engineering with mammalian display by other research groups include: 'Development of a novel mammalian display system for selection of antibodies against membrane proteins (JBC 2020, 295(52):18436-18448)' or 'Antibody-guided design and identification of CD25-binding small antibody mimetics using mammalian cell surface display (Scientific Reports 2021, 11(1):22098). At this stage, my recommendation is minor revision: I would recommend that the authors cite the three papers and additionally more up-to-date research articles focused on antibody engineering through mammalian display and emphasize the novelty of this research, especially in the introduction and discussion sections.

Comments and Suggestions:

1. p6, line 8: the authors states that the presence of 5% BSA makes the survival rate of TR cells at 42 C degree reached 80% and this met their requirement for the antibody evolution. The reason why the presence of 5% BSA simply augmented the survival rate may be discussed in the discussion section. In addition, from what criteria does 80% meet their requirement? This should be stated properly.

2. p7, line 1-2: the author stated that the mutant antibody is more stable because the antibody expression level depends on its stability. This statement may not be true, as there are numerous factors that can influence antibody expression. 

3. The authors did not present the KD values of A115V variant, which only shows increase in the affinity to TNF-alpha but not in the stability in Table 1. Please present the values.

4. Minor grammatical issues or typos (This manuscript needs English editing):
p6, line 36 and 37: 5 types --> 5 different variants, 1 type --> only one variant
p6, line 37-39: These results suggest that the high incubating and labeling temperatures are much more challenging conditions through which newly generated mutants can survive -->
These results suggest that incubating and labeling at higher temperatures make newly generated mutants more challenging to survive.
p6, line 42: Labeling at 4 C degree --> Being incubated at 4 C degree
p6, line 53: scfvs --> scFvs
p7, line 18: acFvs --> scFvs
p7, line 21: dimmers --> dimers
p7, line 39: kappa CL scFv --> Ckappa or kappa chain of CL

5. Supplement data includes inappropriate journal name 'Applied Microbiology and Biotechnology'.

Author Response

Dear Reviewer,

We have tried hard to address all the questions and concerns raised by you on our manuscript titled “Simultaneous maturation of antibody stability and affinity by CHO cell display”. We have revised our manuscript extensively. The added and modified contents in the revised manuscript have been marked in red. I hope that our answers to the questions and the revised manuscript meet your requirements for its publication on bioengineering.

Sincerely yours,

Haiying Hang

Comments and Suggestions for Authors

The research article by Luo et al. titled ‘Simultaneous Maturation of Antibody Stability and Affinity by CHO Cell Display' aims to present antibody maturation methodology to achieve both affinity and stability, utilizing CHO cell display.

This field is a very popular field with multiple research papers published yearly. Although Luo and Hang’s current article is focused on optimization of both the interaction and stability of antibodies against antigens, the same authors published an article on this topic in May 2020 titled ‘High efficiency CHO cell display-based antibody maturation’ (Scientific Reports 2020, 10:8102). The aim of their previous research was to achieve affinity maturation of antibodies using CHO cell display. This topic is very similar to their current article and from the abstract, it can be deducted that the uniqueness in the current paper is the simultaneous maturation of antibody affinity as well as thermostability. Although the authors presented stability issues of the engineered antibodies, there is a lot of overlap of information between the two papers, both covering antibody maturation using AID as a diversity generator.

In addition, competing papers, regarding antibody engineering with mammalian display by other research groups include: 'Development of a novel mammalian display system for selection of antibodies against membrane proteins (JBC 2020, 295(52):18436-18448)' or 'Antibody-guided design and identification of CD25-binding small antibody mimetics using mammalian cell surface display (Scientific Reports 2021, 11(1):22098). At this stage, my recommendation is minor revision: I would recommend that the authors cite the three papers and additionally more up-to-date research articles focused on antibody engineering through mammalian display and emphasize the novelty of this research, especially in the introduction and discussion sections.

Comments and Suggestions:

  1. p6, line 8: the authors states that the presence of 5% BSA makes the survival rate of TR cells at 42 C degree reached 80% and this met their requirement for the antibody evolution. The reason why the presence of 5% BSA simply augmented the survival rate may be discussed in the discussion section. In addition, from what criteria does 80% meet their requirement? This should be stated properly.

We have added the following in Result:"In the labeling process, we want to minimize the loss of cells, so that ensure the diversity of the mutation library. BSA is usually used as a stabilizer to protect protein activity under adverse conditions such as heating. It is also often used as protein supplement in cell culture media. We speculate that the addition of BSA may make the cell membrane and intracellular proteins more stable, thus improving the survival rate of cells."

  1. p7, line 1-2: the author stated that the mutant antibody is more stable because the antibody expression level depends on its stability. This statement may not be true, as there are numerous factors that can influence antibody expression.

The reviewer is correct. Our statement is not logically rigorous. We have changed the statement as following:" hinting that the thermo-stability might be an important factor to guarantee high level expression of this antibody."

  1. The authors did not present the KD values of A115V variant, which only shows increase in the affinity to TNF-alpha but not in the stability in Table 1. Please present the values.

Indeed, we did not present the KD values of A115V obtained through a procedure at normal temperature. We demonstrated that a maturation procedure at normal temperature using CHO cell display were able to obtain mutant antibodies with increased affinities in previous publication. In this study, we focused on the core question whether the our current procedure can simultaneously improve both affinity and thermo-stability.

  1. Minor grammatical issues or typos (This manuscript needs English editing):

p6, line 36 and 37: 5 types --> 5 different variants, 1 type --> only one variant

p6, line 37-39: These results suggest that the high incubating and labeling temperatures are much more challenging conditions through which newly generated mutants can survive --> These results suggest that incubating and labeling at higher temperatures make newly generated mutants more challenging to survive.

p6, line 42: Labeling at 4 C degree --> Being incubated at 4 C degree

p6, line 53: scfvs --> scFvs

p7, line 18: acFvs --> scFvs

p7, line 21: dimmers --> dimers

p7, line 39: kappa CL scFv --> Ckappa or kappa chain of CL

We have revised them in the revised manuscript.

  1. Supplement data includes inappropriate journal name 'Applied Microbiology and Biotechnology'.

We have corrected this mistake in the revised manuscript.

Round 2

Reviewer 3 Report

My review comments are addressed. 

Author Response

Dear reviewer,

Our manuscript has been reviewed by a native English-speaker and revised to improve readability. The revision contents in the revised manuscript have been marked in red. Thank you again for your positive comments and valuable suggestions to improve the quality of our manuscript. 

Sincerely yours,

Haiying Hang